# SPFNO: Spectral operator learning for PDEs with Dirichlet and Neumann boundary conditions

## Abstract

Neural operator has been validated as a promising deep surrogate model for solving partial differential equations (PDEs). Based on the spectral operator learning (SOL) architecture, an enhanced orthogonal polynomial neural operator that we have developed significantly improved the method's accuracy by precisely satisfying the boundary conditions (BCs), but is associated with Gauss-type grids, limiting comparisons on most public datasets. In this paper we introduce SPFNO, a novel SOL method, to learn the target operators on uniform grid datasets for PDEs with non-periodic BCs. Numerical results for various PDEs such as viscous Burgers' equation, Darcy flow and coupled Allen–Cahn equations demonstrate the computational efficiency, resolution invariant property, and BC-satisfaction behaviour of proposed model. An accuracy improvement of approximately 1.7X–4.7X over the non-BC-satisfying baseline is also achieved. Furthermore, studies on SOL emphasize the importance of satisfying BCs as a criterion for deep surrogate models of PDEs.

neural operator, deep learning-based PDE solver, AI4science, scientific machine learning, spectral method

## 1 Introduction

Partial Differential Equations (PDEs) play a pivotal role in various scientific and engineering fields, modeling phenomena such as heat conduction, fluid flow, electromagnetic waves, and quantum mechanics. Given that a substantial majority of PDEs lack analytical solutions, numerical methods, such as spectral methods and finite difference methods, are the primary means of numerically solving PDEs. Recently, researchers have discovered that deep-learning methods can serve as alternatives to these traditional methods. At present, two primary deep learning methodologies are employed for solving PDEs, including directly approximating of the solution using neural networks, e.g., the deep Ritz methods (Yu et al., 2018) and physics informed neural networks (Raissi et al., 2019); or approximating the nonlinear operator between the input and output functions, which is known as the neural operator (Lu et al., 2021; Cai et al., 2021) and the focus of this paper.

The Fourier neural operator (FNO, Zongyi et al. (2021)) is a popular neural operator with applications various fields (Pathak et al., 2022; Zhang et al., 2022; Grady II et al., 2022), followed by many derivative neural operators that adopted a similar spectral analysis backbone and replaced its Fourier transform. We will introduce them in more detail in Sec. 2.1. Meanwhile, there have been multiple studies on the application of the well-known transformer models in solving PDEs (Cao, 2021; Liu et al., 2022a), among which the recently developed latent spectral model (LSM, Wu et al. (2023)) ranked 1st in solving multiple PDE datasets. Thus, the transformers are significant challengers to neural operators in solving PDEs.

The boundary condition (BC) of PDE plays a crucial role in defining the behavior of the system and the treatment of them is the most crucial issue for spectral methods. However, the majority neural operators with spectral structure cannot strictly satisfy BCs that are most commonly used, such as the Dirichlet, Neumann, and Robin BCs (also known as the first-, second-, and third-type BCs, respectively). By generalizing the backbone architecture of FNO using the technique of spectral

methods with a suitable basis, Liu et al. (2022b) introduced a general framework named spectral operator learning (SOL), in which the enhanced neural operators satisfy the BCs exactly. And as the first SOL instance, the orthogonal polynomial neural operator (OPNO) showed state-of-the-art performance on solving PDEs with non-periodic BCs. Moreover, when solving the heat transfer equation with Robin BCs, it acheived an unprecedent relative $L^2$ norm error of $1e-6$ in the implementation of all neural operators. In addition, based on similar ideas, Bonev et al. (2023) developed developed a spherical fourier neural operators (SFNO) that strictly satisfies behavioral boundary conditions on the sphere (Boyd, 2001); while the Boundary enforcing Operator Network (BOON, Saad et al. (2023)) enforce the BCs to arbitrary neural operator using a refinement procedure.

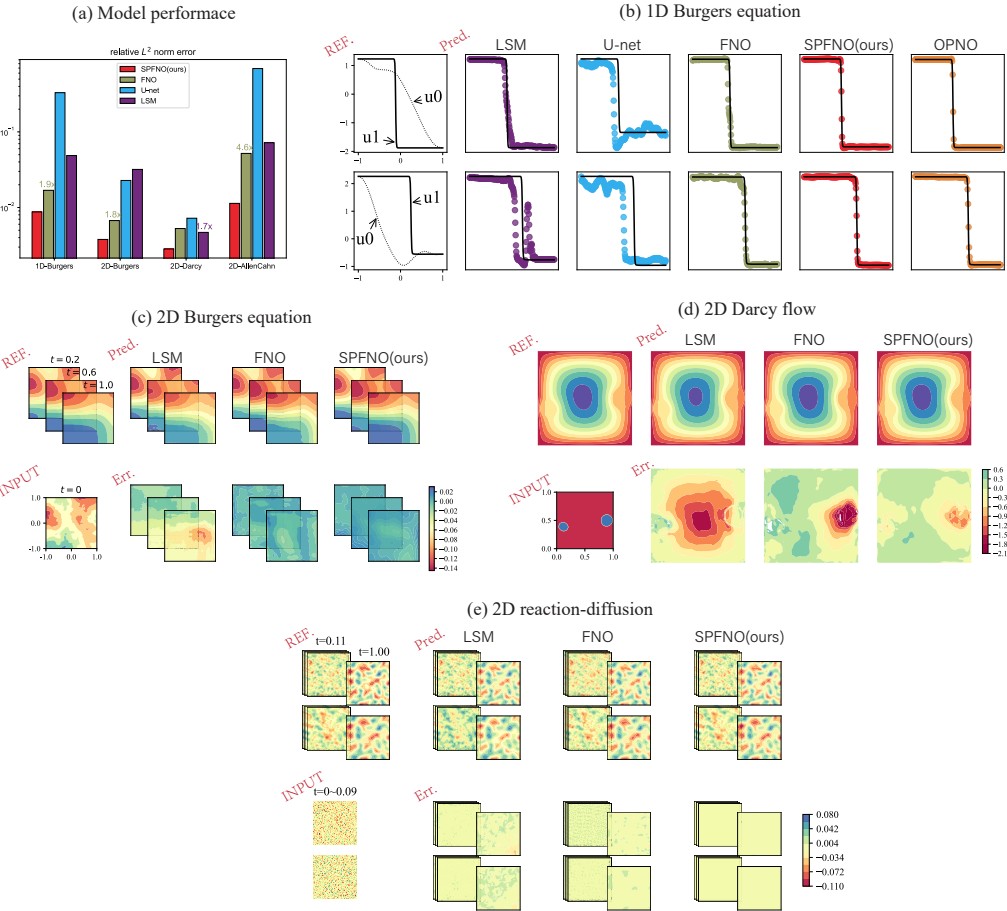

Figure 1: (a) Comparison between SPFNO and non-BC-satisfying baseline models. The lower, the better. (b)–(e) Examples of numerical experiments.

Unfortunately, the fast transformation algorithm of OPNO depends on the Gaussian grids, while only values on uniform grids are given in the majority of public datasets, limiting the comprehensive comparison between SOL architecture with non-BC-satisfying models. To address this technical issue, we introduce a novel SOL method, named SPFNO, of which the **fast transformation algorithm** with a time complexity of $O(NlogN)$ is designed on a $N$-point **uniform grid**, to solve PDEs with Dirichlet and Neumann BCs. The specified trigonometric bases allow the errors on the BCs to reach machine precision. In addition, SPFNO also possesses the following appealing properties that are expected from a neural operator.

- **Invariance to discretization**. Without the need for retraining, an SPFNO model trained on a coarse grid can directly predict the solution on any fine grid. Detailed discussion on this property is conducted in Sec. 3.5.2.

- **Efficient and accurate spatial differentiation**. Differentiating the output function of SPFNO requires operations of only $O(NlogN)$ using the spectral method.

Comparisons of the performance and several examples are given in Fig. 1. All datasets used are already public, and the code and pre-trained model are made publicly available at `https://github.com/lab-nameless/sol`.

## 2 METHODOLOGY

### 2.1 SPECTRAL OPERATOR LEARNING

The SOL is a kind of specifically designed neural architecture that consists of spectral analysis and linear learnable transformation and, at the same time, it reveals the concept of strictly satisfying boundary conditions in deep surrogate models for PDEs. It allows for the utilization of various spectral methods techniques in deep learning methods and leads to improved accuracy and credibility of the neural operator models.

We now briefly introduce the SOL architecture (Liu et al., 2022b) through the task of prediction of time-dependent PDEs. Since its application is not limited to this scenario, other cases will be discussed in the numerical experiments. Consider the following PDE

$$u_t(x,t) + \mathcal{N}(u(x,t)) = 0, x \in \Omega, t \in (0,T]$$

with inital–boundary conditions

$$\begin{aligned} u(x,0) &= u_0(x), x \in \Omega, \\ \mathcal{B}(u(x,t)) &= 0, x \in \partial\Omega, \end{aligned} \tag{1}$$

where $\mathcal{N}$ is a continuous operator and $\mathcal{B}$ is an operator corresponding to specific BCs. The task is to learn the solution operator $\mathcal{S}_\tau$ which evolves the initial condition $u_0$ to the solution at $\tau$, namely, $\mathcal{S}_\tau(u_0(x)) = u(x,\tau)$. We let $\{u_k(x)\}_{k\in\mathbb{N}}$ and $\mathcal{T}$ be a set of basis functions verifying the BCs (1) and the linear transform induced by such a basis, respectively. Then the SOL model for $\mathcal{S}(\tau)$ consists of a stack of neural spectral layers $\mathcal{L}^{(l)}$ that are induced by $\{u_k\}$ and $\mathcal{T}$, and are in the form of

$$u^{(l+1)} = \sigma(W_l u^{(l)} + \mathcal{L}^{(l)} u^{(l)}) := \sigma(W_l u^{(l)} + \mathcal{T}^{-1} A_l \mathcal{T} u^{(l)}),$$

where $\sigma$ is the nonlinear activation function, $W_l$ is an auxiliary pointwise shallow neural network, and $A_l$ is a learnable spectrum-wise matrix. This architecture is first demonstrated in Zongyi et al. (2021) for FNO, and then adopted by multiple neural operators, such as the multiwavelet-based neural operator (MWT-NO, Gupta et al. (2021)), wavelet neural operator (WNO, Tripura & Chakraborty (2023)), spectral neural operator (SNO, Fanaskov & Oseledets (2022)), OPNO (Liu et al., 2022b), multi-channel IAE-Net (Ong et al., 2022) and so on. Examples are listed in Tab. 1.

Table 1: The fundamental spectral bases, the specified BCs and the types of utilized grids of neural operators that are based on spectral analysis. Only a subset of models are listed.

| Model | FNO | MWT-NO | WNO | IAE-Net | SFNO | SOL | |
| --- | --- | --- | --- | --- | --- | --- | --- |
| | | | | | | OPNO | SPFNO |
| Basis | Fourier | multiwavelets | wavelets | integral | spherical harmonics | Shen polynomials | trigonometric polynomials |
| BCs | periodic (implicitly) | — — | | | spherical | Dirichlet Neumann Robin | Dirichlet Neumann |
| Grid type | uniform | | | | spherical coordinates | Gaussian | uniform |

### 2.2 SPFNO

In this subsection, the domain $\Omega$ is limited to one dimensional interval $[0,1]$ for convenience, while all the conclusions can be easily generalized to separable multi-dimensional domains. When we focus the discussion on the cases of Dirichlet or Neumann BCs, one possible choice in the numerical method is the basis of specified trigonometric polynomials with "semi-period" and corresponding

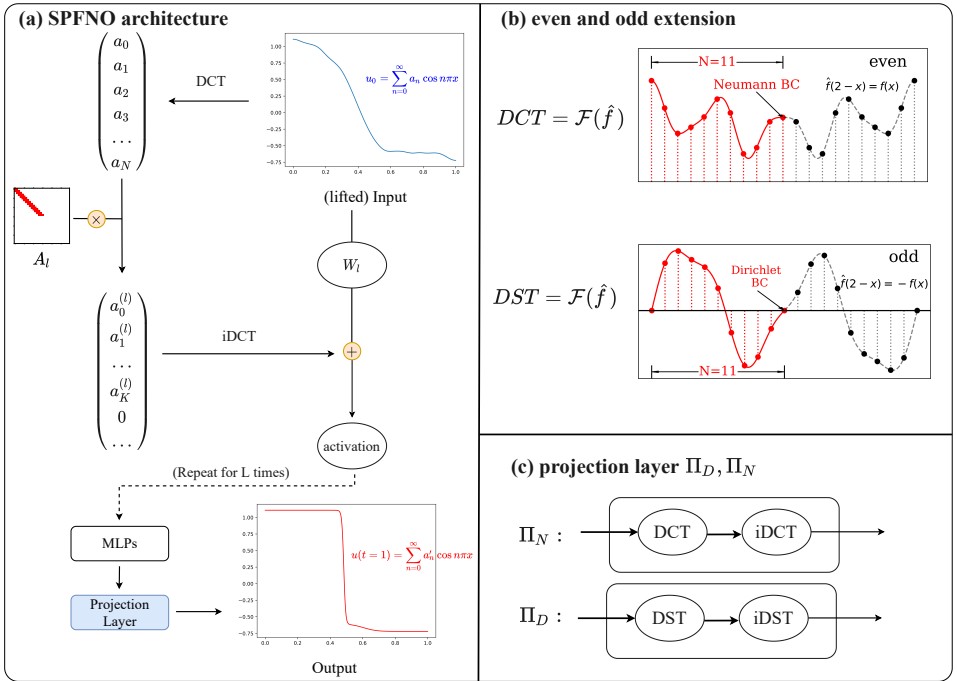

Figure 2: Sketch map of SPFNO

transformations, i.e., $u_k(x) = \sin k\pi x, x \in [0, 2]$ and (discrete) sine transform for Dirichlet BCs; or the basis of $u_k(x) = \cos k\pi x, x \in [0, 2]$ and (discrete) cosine transform for Neumann BCs. Such neural operators are named SPFNO, which is short for the Semi-Periodic FNO and SPecified FNO with non-periodic boundary conditions.

The spectral methods of the aforementioned specified trigonometric polynomials have been investigated in Bueno-Orovio et al. (2014). SPFNO is a natural derivative of these methods under the framework of SOL. For convenience, the operators associated with the sine and cosine bases are denoted by SinNO and CosNO, respectively.

The famous discrete sine/cosine transforms (DST/DCT) of the 1st kind are based on odd/even extensions, respectively, where the solutions are discretized at uniform grids and BCs are imposed the on boundary points, see Fig 2 (b). With a slight abuse of notation, we denote $\hat{f}$ as the odd extension of function $f$ if $f$ satisfies Dirichlet BCs or the even extension if $f$ satisfies Neumann BCs:

$$\text{Odd: } \hat{f}(x) = \begin{cases} f(x), \ x \in [0, 1] \\ -f(2 - x), \ x \in [1, 2] \end{cases}; \quad \text{Even: } \hat{f}(x) = \begin{cases} f(x), \ x \in [0, 1] \\ f(2 - x), \ x \in [1, 2] \end{cases}.$$

Then, the specified trigonometric polynomials form a basis for $\left\{ \hat{f} | f \in C(\Omega) \right\}$.

**Theorem 2.1.** *The extended function $\hat{f}$ can be uniquely deconstructed by cosine polynomials $\{\cos kx\}_{k \in N}$ if and only if $f \in C[0, 1]$ satisfies the Neumann BCs and can be uniquely deconstructed by sine polynomials $\{\sin kx\}_{k \in N+}$ if and only if $f$ satisfies the Dirichlet BCs.*

The proof is provided in Sec. A.1. This theorem ensures the effectiveness of DST and DCT as decomposition transforms in solving PDEs with corresponding specific BCs. Finally, the structure of the SPFNO is given in Fig. 2. Theorem 2.1 also leads to the following conclusion:

**Theorem 2.2.** *The outputs of SinNO and CosNO strictly satisfy the Dirichlet and Neumann BCs, respectively.*

## 3 NUMERICAL EXPERIMENTS

In order to verify the accuracy and efficiency of the proposed SPFNO method and the importance of BC-satisfying property for neural operators, we compare it with multiple popular but non-BC-satisfying architectures by solving the following four PDEs on publicly available datasets: (1) 1-D and 2-D Burgers' equations with Neumann BCs; (2) Darcy flow problem with Dirichlet BCs; and (3) the coupled reaction diffusion equations with Neumann BCs. The accuracy of the models is measured by the average relative $L^2$ norm error (also known as the relative mean square error, or RMSE) between the predicted solution and the reference solutions and the $L^\infty$ norm error on the corresponding BCs. The maximum $L^2$ norm error on the test dataset, which represents the empirical worst performance and is crucial for assessing the credibility of models, is also taken into consideration. The baseline models are listed below.

1. **FNO** (Zongyi et al., 2021) is a state-of-the-art neural operator for parametric PDEs, especially those involving periodic BCs. FNOs have many applications and they have achieved impressive accuracy in practical application due to its architecture being similar to that of the spectral method.

2. **OPNO** (Liu et al., 2022b) is the first proposed SOL method for non-periodic BCs such as Dirichlet, Neumann, and Robin BCs. It provided the first numerical example that verifies the competitive accuracy of deep-learning-based surrogate model to the numerical method, with the relative errors reaching the order of $10^{-6}$. The OPNO will be tested if and only if the data on Chebyshev-Gauss-Lobatto grids are provided.

3. **U-net** (Ronneberger et al., 2015) is a popular autoencoding deep learning architecture that combines the convolutional and deconvolutional layers. It has been proven to be a powerful model for tackling image segmentation tasks. It is used as a baseline model in the PDEBench datasets and demonstrates considerable accuracy in specific tasks.

4. **LSM** (Wu et al., 2023) is a cutting-edge transformer-based neural PDE solver that consists of an autoencoding backbone and innovative neural spectral blocks. It successfully trained a neural network with considerable depth and outperformed the performance of $14$ existing models, including neural operators, autoencoders, and transformers, across 7 PDE-solving tasks. In this paper, the LSM model also serves as the most accurate non-neural-operator model for PDEs known to us.

5. **BOON** (Saad et al., 2023) is a method that enforces the BCs on arbitrary neural operators by making structural changes to the operator kernel. It proved that the BC-satisfying correction significantly increase the accuracy of the neural operators.

The comparison, however, is not limited to the aforementioned models. Since the datasets utilized in the experiments are publicly available, readers may refer to the corresponding papers and their subsequent articles for the results of other models.

Except in Sec. 3.5.2, all models are evaluated at the same resolution as that used during the training procedure. All experiments are performed on an Nvidia A100 GPU. For fairness, all models are trained for the same number of epochs and with random seeds fixed to 0.

### 3.1 EXAMPLE 1: 1D VISCOUS BURGERS' EQUATION WITH NEUMANN BCS

We first consider the one-dimensional viscous Burgers equation as follows

$$\partial_t u(x,t) + \frac{1}{2}\partial_x(u^2(x,t)) = \nu\partial_{xx}u(x,t), x \in \Omega \tag{2}$$

subject to the initial-boundary conditions

$$u(x,0) = u_0(x,t), \ x \in \Omega, \tag{3}$$

$$\frac{\partial u}{\partial \mathbf{n}}(x,t) = 0, \ x \in \partial\Omega, \tag{4}$$

where $\Omega = [-1,1]$. We are interested in learning the solution operator $S_1 : u_0(x) \to u(x,1)$. Burgers' equation is a fundamental PDE with applications in modeling turbulence, nonlinear acoustics, and traffic flow. The complexity of the dynamical system it describes poses challenges for the

Table 2: Evaluation of the relative $L^2$ norm error ($\times 10^{-2}$), and worst error ($\times 10^{-2}$), and the error on the Neumann BC of 1D-Burgers' equation with resolution $N$

(a) Relative $L^2$ norm error and worst error

| | $N = 256$ | | $N = 1024$ | | $N = 4096$ | |
| | $L^2$ | worst | $L^2$ | worst | $L^2$ | worst |
|---|---|---|---|---|---|---|
| FNO | 1.57 | 14.9 | 1.68 | 14.1 | 1.69 | 16.6 |
| Unet | 6.27 | 56.9 | 27.6 | 126.3 | 33.0 | 146.3 |
| LSM | 4.87 | 47.2 | 21.1 | 108.6 | 38.7 | 167.7 |
| BOON | 1.20 | 10.1 | 1.28 | 10.2 | 1.42 | 10.0 |
| OPNO | **0.770** | *4.63* | **0.781** | *4.40* | **0.782** | *3.86* |
| CosNO | *0.868* | **3.58** | *0.862* | **3.55** | *0.873* | **3.65** |

(b) Error on the BC

| | $N = 256$ | $N = 1024$ | $N = 4096$ |
| | $L^\infty_{BC}$ | $L^\infty_{BC}$ | $L^\infty_{BC}$ |
|---|---|---|---|
| FNO | 2.9$E$-1 | 4.1$E$-1 | 5.5$E$-1 |
| Unet | 1.1$E$-2 | 5.7$E$-2 | 8.9$E$-2 |
| LSM | 4.8$E$-1 | 5.1$E$-1 | 9.8$E$-1 |
| BOON | 0 | 0 | 0 |
| OPNO | 6.0$E$-12 | 1.1$E$-10 | 1.9$E$-9 |
| CosNO | 0 | 0 | 0 |

learning of deep models, so it has been adopted as one of the most popular benchmark problems in the field of AI4Science.

As Liu et al. (2022b) concluded, the BC-satisfying property is crucial for enhancing the accuracy and credibility of surrogate models for PDEs. If the statement holds, the SPFNO should also exhibit prominent superiority, even when employed on different grids, since the SOL architecture is designed to learn the operator.

We adopt CosNO to match the Neumann BCs and the same quasi-diagonalizing techniques for the learnable spectral matrix $A_l$ as in Liu et al. (2022b), i.e., for the 1D Burgers equation, we set the bandwidth of $A_l$ to 4. Such a technique is not suitable for FNO, so the bandwidth of FNO is maintained at 1, see Fig. 4. The results illustrated in Tab. 2 and Fig. 4 show that the SPFNO can reach a comparable accuracy compared with the SOTA model OPNO, and thus remarkably outperforms all non-BC-satisfactory models. The relative $L^2$ errors are reduced by an approximately 45.0%, and the maximum errors are much lower, which suggests more reliable predictions.

## 3.2 EXAMPLE 2: 2D VISCOUS BURGERS' EQUATION WITH NEUMANN BCS

With the computational domain set to $\Omega = [-1, 1]^2$, the 2-D Burgers' equation (10) − (12) is considered in this experiment. Nevertheless, the output of the target operator consists of solutions at multiple fixed time instances, i.e. $S_{\tau_1,...,\tau_n} : u_0 \mapsto \{u(\cdot, \tau_1), ..., u(\cdot, \tau_n)\}$, so that the time-dependent PDEs can be efficiently solved by only one forward propagation. Moreover, both the operator and the task can be more complicated compared with the 1-D case. We choose a subset of time discretization by fixing $\{\tau_i\}_{i \leq n} = \{0.2, 0.6, 1.0\}$. During the training, the bandwidth of the learnable matrix $A_l$ is set as 4. The results are illustrated in Tab. 3, where the results of different SOL models are very close, and achieve the state-of-the-art performance among tested models.

Table 3: Evaluation of the relative $L^2$ norm error ($\times 10^{-2}$) and worst error ($\times 10^{-2}$) of 2D-Burgers' equation with resolution $N \times N$

| | $N = 50$ | | $N = 100$ | | $N = 200$ | |
| | $L^2$ | worst | $L^2$ | worst | $L^2$ | worst |
|---|---|---|---|---|---|---|
| FNO | 0.528 | 9.02 | 0.589 | 10.03 | 0.672 | 9.71 |
| Unet | 1.64 | 16.0 | 2.31 | 17.6 | 2.28 | 17.1 |
| LSM | 2.43 | 9.23 | 2.93 | 12.4 | 3.19 | 11.6 |
| OPNO | **0.371** | **3.37** | **0.336** | **3.68** | **0.335** | **3.68** |
| CosNO | *0.386* | *4.98* | *0.378* | *5.00* | *0.378* | *5.05* |

## 3.3 EXAMPLE 3: COUPLED 2D REACTION–DIFFUSION EQUATIONS WITH NEUMANN BCS

The coupled reaction–diffusion (Allen Cahn) equations are formulated as follows

$$\frac{\partial u}{\partial t} = d_u \frac{\partial^2 u}{\partial x^2} + d_u \frac{\partial^2 u}{\partial y^2} + R_u(u, v), \ x \in (-1, 1)^2,$$

$$\frac{\partial v}{\partial t} = d_v \frac{\partial^2 v}{\partial x^2} + d_v \frac{\partial^2 v}{\partial y^2} + R_v(u, v). \ x \in (-1, 1)^2,$$

where

$$R_u(u,v) = u - u^3 - k - v,$$
$$R_v(u,v) = u - v,$$
$$d_u = 0.001, d_v = k = 0.005.$$

The nonlinearly coupled variables $u$ and $v$ represent the activator and inhibitor in the system, respectively, to which the Neumann BCs are imposed. The dataset is provided by PDEBench (Takamoto et al., 2022), a comprehensive set of benchmarks for scientific machine learning. Since the data are given on a staggered uniform grid, directly sub-sampling would yields a non-uniform grid. So we only perform the experiment with the original resolution of $128 \times 128$. The bandwidth of SPFNO is fixed to 1.

According to the settings of the PDEBench dataset , models are trained using an autoregressive approach. A similar approach is often applied in predicting weather with neural operators (Pathak et al., 2022). The results are shown in Tab. 5, where SPFNO acheives the lowest errors.

Table 4: relative $L^2$ norm error ($\times 10^{-2}$) and worst error ($\times 10^{-2}$) of 2D reaction diffusion equation

|  | $L^2$ error | worst error | #param | #time(s) |
|---|---|---|---|---|
| FNO | *5.19* | *6.37* | 1.3m | 150 |
| Unet | 68.9 | 77.6 | 7.8m | 217 |
| LSM | 7.20 | 13.7 | 1.2m | 606 |
| CosNO | **1.13** | **1.60** | 1.4m | 275 |

## 3.4 EXAMPLE 4: 2-D DARCY FLOW WITH DIRICHLET BCS

Darcy's law describes the flow of fluid through a porous medium. It has been widely implemented in various fields, including hydrogeology, petroleum engineering, and soil mechanics. In this experiment, the 2-D steady-state Darcy flow equations in a unit box are formulated as the following boundary value problem (BVP):

$$-\nabla \cdot (a(x)\nabla u(x)) = f, x \in [0,1]^2. \tag{5}$$

Moreover, the homogeneous Dirichlet BCs are imposed. The task is to learn the operator $\mathcal{G}(a) = u$ that maps the diffusion coefficient to the solution. This problem serves as another most commonly used benchmark for deep PDE solvers since the dataset is provided in Zongyi et al. (2021). In this dataset, the diffusion coefficient $a(x)$ is taken as a piecewise constant, while the reference solution is generated using a finite difference method. Under this premise, however, the nondifferentiable variable coefficient makes the 2nd order finite difference method unsuitable for solving the problem. So we utilize the 2D-Darcy dataset with $f$ fixed as 100.0 in Takamoto et al. (2022) instead. Its larger size ($10^4$ pieces of data compared to $10^3$ in Zongyi et al. (2021)) also contributes to the model achieving higher accuracy.

Compared to the Neumann BC, the Dirichlet BC is much easier to learn because it does not involve any derivative. Additionally, the heterogeneity of the input and output functions leads to a much more complicated spectral structure of the mapping operator. Actually, multiple non-neural-operator method have been reported to surpass the performance of neural operators in solving Eq. (5), especially the transformers, among which the LSM acheives the highest accuracy that is known to us.

We adopt the ReduceLROnPlateau scheduler for SPFNO to accelerate the training. The results can be found in Tab. 5, where the SinNO again obtains the lowest relative error, while the LSM notably achieves the lowest maximum error.

## 3.5 ADDITIONAL EXPERIMENTS

### 3.5.1 EVALUATION AND COMPARISON ON THE DATASET FROM BOON

We additionally compare our model with the BOON model on the Dirichlet and Neumann datasets that the latter provided. For details on the generation of the dataset, please refer to Saad et al. (2023).

Table 5: Evaluation relative $L^2$ norm error ($\times 10^{-2}$) and worst error ($\times 10^{-2}$) for 2D Darcy flow problem

|       | $L^2$ error | worst error | #param | #time |
|-------|-------------|-------------|--------|-------|
| FNO   | 0.688       | 6.04        | 2.4m   | 7.44  |
| Unet  | 0.989       | 4.81        | 7.8m   | 6.53  |
| LSM   | *0.468*     | **2.72**    | 19.2m  | 62.7  |
| SinNO | **0.283**   | 4.89        | 2.4m   | 32.9  |

To accelerate the training process, a ReduceLROnPlateau scheduler is adopted as in Example 4 for the SPFNO model.

**(1) 1D Burgers' equation with Dirichlet BCs:** In this problem and dataset, the following viscous Burgers' equation with Dirichlet BCs is consider:

$$u_t + (u^2/2)_x = \nu u_{xx}, \; x \in [0,1], t \geq 0,$$
$$u_0(x) = \begin{cases} u_L, \text{ if } x \leq 0.5, \\ u_R, \text{ if } x > 0.5, \end{cases}$$
$$u(0,t) = u_{\text{exact}}(0,t), \; u(1,t) = u_{\text{exact}}(1,t), \; t > 0.$$

The task is to learn the solution operator $S : u_0(x) \to u_{t=1.2}$.

**(2) 2D wave equation with Neumann BCs:** In this problem and dataset, the following 2D wave equation with Neumann BCs is consider:

$$u_{tt} = c^2(u_{xx} + u_{yy}), \; x,y \in [0,1]^2, \; t \geq 0,$$

subject to homogeneous Neumann BCs, to which the following analytical solution exists

$$u_{\text{exact}}(x,t) = k\cos(\pi x)\cos(\pi y)\cos(c\sqrt{2}\pi t).$$

The task is to learn the solution operator $S : u_0(x,y) \to \{u(x,y,t=t_M)\}$.

The results are illstrated in Tab. 6 and 7, respectively, where the SPFNOs outperform BOONs in almost all of the cases. It is worth noting that, however, we believe that the primary factor contributing to the extremely high accuracy of SPFNO is the dataset. The problem has an analytical solution, and the generation of input functions involves only a limited number of degrees of freedom. This circumstance enables the model to easily fit a manifold of significantly reduced dimensionality while learning the solution operator. Hence, the existing results sufficiently illustrate the approximation capability of all tested model for the problems. However, the practical implications of further error reduction remain limited.

Table 6: Single-step prediction for 1D Burgers' with Dirichlet BC. Relative $L^2$ test error(error on BCs) for Burgers' equation with varying viscosities $\nu$ at resolution $N = 500$

| Model | $\nu = 0.1$ | $\nu = 0.05$ | $\nu = 0.02$ | $\nu = 0.005$ | $\nu = 0.002$ | #Time |
|-------|-------------|--------------|--------------|---------------|---------------|-------|
| SinNO (ours) | **1.12e − 5**(0) | **1.26e − 5**(0) | **5.95e − 5**(0) | **5.04e − 4**(0) | 7.28e − 4(0) | 0.12s |
| BOON-FNO (Saad et al., 2023) | 1.2e − 4(0) | 1.0e − 4(0) | 8.4e − 5(0) | 1.0e − 4(0) | 1.27e − 3(0) | 1.3s |
| BOON-MWT (Saad et al., 2023) | 2.0e − 4(0) | 2.5e − 4(0) | 2.2e − 4(0) | 2.0e − 4(0) | **3.4e − 4**(0) | − − |

Table 7: Multi-step prediction for the 2D wave equation with Neumann BC. Relative $L^2$ error (error on BCs) for various benchmarks with varying resolutions N and M = 25.

| Model | $N = 25$ | $N = 50$ | $N = 100$ |
|-------|----------|----------|-----------|
| CosNO | **1.14e − 4**(0) | **4.69e − 5**(0) | **1.93e − 4**(0) |
| BOON (Saad et al., 2023) | 9.7e − 4(0) | 8.93e − 4(0) | 9.6e − 4(0) |

### 3.5.2 THE EVALUATION OF THE PRE-TRAINED MODEL ON DIFFERENT GRID

We take the pre-trained model in Experiment 1 as an example. As a consequence of the spectral structure of the SOL architecture, the model trained on the coarse grid ($N = 256$) can directly predict on the fine grid without significant loss in accuracy, see Tab 8 and Fig. 3. The data from the sub-scaled grid remains untouched during the training process.

Table 8: The evaluation relative $L^2$ errors ($L^\infty$ errors on BCs) of SPFNO model for 1D Burgers' equation. Model is trained on a $N$ grid but evaluated on a grid with resolution $N'$

|  | $N' = 256$ | $N' = 512$ | $N' = 1024$ | $N' = 2048$ | $N' = 4096$ |
|---|---|---|---|---|---|
| $N = 256$ | 0.0086766(**0**) | 0.0086748(**0**) | 0.0086737(**0**) | 0.0086732(**0**) | 0.0086729(**0**) |
| $N = 4096$ | 0.0087394(**0**) | 0.0087364(**0**) | 0.0087351(**0**) | 0.0087344(**0**) | 0.0087341(**0**) |

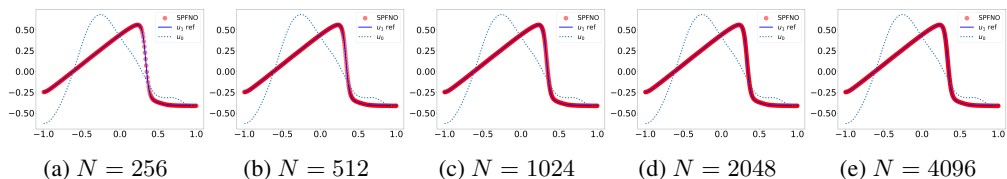

(a) $N = 256$    (b) $N = 512$    (c) $N = 1024$    (d) $N = 2048$    (e) $N = 4096$

Figure 3: The prediction of SPFNO model that is trained on a $N = 256$ grid but evaluated on different grids

## 4 DISCUSSION AND FUTURE WORK

We presented a novel spectral operator learning (SOL) architecture for PDEs with Dirichlet and Neumann BCs. This method, named SPFNO, combines traditional spectral methods and the neural operator architecture, so that it satisfies the corresponding BCs exactly. The BC-satisfying properties were proved both theoretically and numerically. Numerical experiments also showed that the SOL methods yield very close results regardless of the different types of grids they are associated with. On the other hand, compared with models including non-BC-satisfying models and BC-satisfying BOON model, state-of-the-art performance in solving a variety of widely adopted benchmark problems can be achieved with our proposed framework. From the perspective of machine learning, the BC-satisfying spectral structure is an intuitive bias that effectively shrinks the hypothesis space.

Although the paper focuses on the data-driven training of neural operators, we can also directly learn the target operator by utilizing the physics constraints and minimizing the residuals of equations, which can reduce the dependence on datasets. Readers may find that the residuals of the BCs that are usually difficult to handle will vanish for an SOL, making the training much easier. In addition, developing other SOL instances for more complex BCs and geometries, e.g., the radiation BCs or unbounded domains, is an important subject of future research.

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

# A  APPENDIX

## A.1  PROOF FOR THEOREM 2.1

Since $f$ is usually a strong solution to some PDEs in the implementation of SPFNO, it is reasonable to assume the smoothness so that at least $f \in H^1([0,1])$. But for now we only need $f \in C[0,1]$, or the differentiability on the boundary additionally for the following proof of Neumann BCs.

### A.1.1  CASE OF $f$ SATISFYING DIRICHLET BCS AND $\hat{f}$ BEING ITS ODD EXTENSION

First, we will prove that if the odd extension $\hat{f}$ can be uniquely deconstructed by sine polynomials, i.e.

$$\hat{f} = \sum_{k=1}^{\infty} b_k \sin k\pi x, \ x \in [0,2],$$

then $f$ satisfies the Dirichlet BCs:
$$f(0) = f(1) = 0.$$
The proof is straightforward: substituting $x = 0$ and $x = 1$ give that
$$f(0) = \hat{f}(0) = 0 = \hat{f}(1) = f(1).$$
On the other hand, given $f \in C[0,1]$ as an arbitrary function that satisfies homogeneous Dirichlet BCs, i.e.,
$$f(0) = f(1) = 0,$$
its odd extension being

$$\hat{f}(x) = \begin{cases} f(x), \ x \in [0,1], \\ -f(2-x), \ x \in [1,2], \end{cases}$$

which means that $\hat{f}$ is a continuous function on a closed interval. Consequently, it can be inferred from the Weierstrass approximation theorem for trigonometric series that the (trigonometric) Fourier series of $\hat{f}$ converges to $\hat{f}$ uniformly, and is of the form

$$\hat{f}(x) = \sum_{m=0}^{\infty} a_m \cos m\pi x + \sum_{n=1}^{\infty} b_n \sin n\pi x, \ x \in [0,2], \tag{6}$$

where

$$a_m = \int_0^2 \hat{f}(x) \cos m\pi x \ \mathrm{dx}, \ \mathrm{m} \in \mathbb{N},$$

$$b_n = \int_0^2 \hat{f}(x) \sin n\pi x \ \mathrm{dx}, \ \mathrm{n} \in \mathbb{N}^+. \tag{7}$$

Then it yeilds that

$$\hat{f}(x) - \sum_{n=1}^{\infty} b_n \sin n\pi x = \sum_{m=0}^{\infty} a_m \cos m\pi x, \ x \in [0,2]. \tag{8}$$

Noting that the left-hand side of Eq. (8) is an odd function while the right-hand side is even. The only possibility is that it remains zero constantly. As a result,

$$\hat{f}(x) = \sum_{n=1}^{\infty} b_n \sin n\pi x,$$

where $b_n$ is determined by Eq. (7).

### A.1.2 Case of $f$ satisfying Neumann BCs and $\hat{f}$ being its even extension

The proof of this part is actually analogous to Sec. A.1.1. Assume that $f \in C[0,1]$ is differentiable on $x = 0$ and 1. On the one hand, given that the even extension $\hat{f}$ can be uniquely deconstructed by cosine polynomials, i.e.

$$\hat{f} = \sum_{k=0}^{\infty} a_k \cos k\pi x, \ x \in [0,2].\tag{9}$$

Noting that $\hat{f}$ is an even expansion. Consequently, we have

$$
\begin{aligned}
f'(1) &= (\hat{f})'(1)\\
&= \lim_{h\to 0} \frac{\hat{f}(1+h) - \hat{f}(1-h)}{2h}\\
&= \lim_{h\to 0} \frac{f(1-h) - f(1-h)}{2h} = 0.
\end{aligned}
$$

Moreover, since Eq. (9) holds, we can further extend $\hat{f}$ to $\mathbb{R}$ with a period of 2, which is denoted as $\tilde{f}$. The periodicity of $\tilde{f}$ yields that the following equation holds on $x = 0$:

$$
\begin{aligned}
f'(0) &= (\tilde{f})'(0)\\
&= \lim_{h\to 0} \frac{\tilde{f}(0+h) - \tilde{f}(0-h)}{2h} = \lim_{h\to 0} \frac{f(0+h) - \hat{f}(2-h)}{2h}\\
&= \lim_{h\to 0} \frac{f(h) - f(h)}{2h} = 0.
\end{aligned}
$$

So the Neumann BCs are satisfied by $f$.

On the other hand, similar to Sec. A.1.1, the Fourier series of $\hat{f}$ converges to $\hat{f}$ uniformly, so Eqns. (6)–(8) also hold for the even extension $\hat{f}$ when $f$ satisfies Neumann BCs. Then it yields that

$$\hat{f}(x) - \sum_{m=0}^{\infty} a_m \cos m\pi x = \sum_{n=1}^{\infty} b_n \sin n\pi x \equiv 0, \ x \in [0,2]$$

As a result,

$$\hat{f}(x) = \sum_{m=0}^{\infty} a_m \cos m\pi x,$$

where $a_m$ is determined by Eq. (7).

**Corollary A.1.** *The sets of specified trignometric functions $\left\{\sqrt{2}\cos k\pi x\right\}_{k\in N}$ and $\left\{\sqrt{2}\sin k\pi x\right\}_{k\in N^+}$ form the orthonormal bases for corresponding function spaces, i.e. the even extension $\left\{\hat{f}_{\text{even}} | f \in C[0,1], \ f'(0) = f'(1) = 0\right\}$ and odd extension $\left\{\hat{f}_{\text{odd}} | f \in C[0,1], \ f(0) = f(1) = 0\right\}$, respectively.*

*Proof.* As Theorem 2.1 has shown that the specified trignometic functions form a basis for corresponding function space, all we need to do is to prove is the orthonormality of the normalized bases. A simple calculation yields

$$\int_0^2 \sin k\pi x \sin m\pi x = -\frac{1}{2}\int_0^2 \cos\pi(k+m)x - \cos\pi(k-m)x = \delta_{k,m},$$

$$\int_0^2 \cos k\pi x \cos m\pi x = \frac{1}{2}\int_0^2 \cos\pi(k+m)x + \cos\pi(k-m)x = \delta_{k,m}.$$

$$\int_0^2 \sin k\pi x \cos m\pi x = \frac{1}{2}\int_0^2 \sin\pi(k+m)x + \sin\pi(k-m)x = 0.$$

In fact, for the normalized trignometic functions $\left\{\sqrt{2}\cos k\pi x\right\}_{k\in N}$ and $\left\{\sqrt{2}\sin k\pi x\right\}_{k\in N^+}$ and unextended function $f \in C[0,1]$ with corresponding BCs, the same process also lead to a similar conclusion.

$\square$

# B    DETAILS OF DATASETS AND NUMERICAL EXPERIMENTS

For neural operators possessing spectral structures, such as those listed in Table 1, the primary parameters determining their approximation capabilities are the width of channels $W$, the depth of the spectral layer $L$, and the number of modes truncated in the spectral operator $(K)$. This criterion can refer to the reference (Boyd, 1978), where the author used the number of modes required for discretizing the same function as a decisive criterion for comparing spectral methods of different bases, including Chebyshev, Fourier, and spherical harmonics. The bandwidth $b$ of the learnable matrix is generally considered to be determined by the orthogonality of the basis. For instance, it does not directly affect the capability of FNO, as illustrated in Fig. 4. However, for SPFNO, a suitable increase in $b$ improves its approximation capability, for which the theoretical analysis is an interesting topic for further research.

Given that all datasets are provided in the references, we adopt to the same parameters of neural operator as stated in the original text for SPFNO. $L$ is fixed to 4 for all experiments. And although different models require significantly different GPU times for training of a single epoch, we train them for the same number of epochs as in the references.

## B.1    EXAMPLE 1: 1D VISCOUS BURGERS' EQUATION WITH NEUMANN BCS

We consider the one-dimensional viscous Burgers equation with Neumann BCs

$$\partial_t u(x,t) + \frac{1}{2}\partial_x(u^2(x,t)) = \nu\partial_{xx}u(x,t), x \in \Omega \tag{10}$$

subject to the initial-boundary conditions

$$u(x,0) = u_0(x,t), \ x \in \Omega, \tag{11}$$

$$\frac{\partial u}{\partial \mathbf{n}}(x,t) = 0, \ x \in \partial\Omega, \tag{12}$$

where $\Omega = [-1,1]$. We are interested in learning the solution operator $S_1 : u_0(x) \to u(x,1)$. In the dataset, the initial condition $u_0(x)$ is generated using a Gaussian random field according to $u_0 \sim \mu$, where $\mu = \mathcal{N}(0, 625(-4\Delta + 25I)^{-2})$ with Neumann BCs, and $\nu$ is fixed to $0.1/\pi$. The dataset consists of 1000 training instance and 100 test instance.

During the training, parameters for SPFNO are chosen as $W = 50$, $K = 20$ and $b = 4$, and all models are trained for 5000 epochs to ensure they are well-trained, following the instruction of Liu et al. (2022b). Besides, we compare the performance of the model width different bandwidth $b$, see Fig. 4.

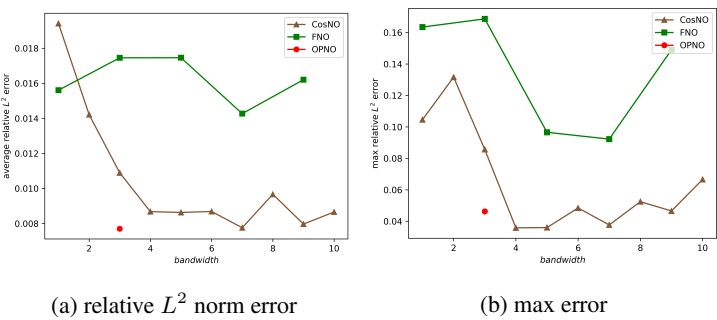

(a) relative $L^2$ norm error          (b) max error

Figure 4: **Comparisons of the relative $L^2$ errors and max errors on the testing data for different $A_l$ bandwidth.** While numerical experiment demonstate that the quasi-diagnolizing technique may improve the performance of SPFNO, it cannot substitute the requirement of BC-satification property, see the results of FNO.

## B.2    EXAMPLE 2: 2D VISCOUS BURGERS' EQUATION WITH NEUMANN BCS

We consider the two-dimensional viscous Burgers equation with Neumann BCs

$$\partial_t u(x,y,t) + (u \cdot \nabla)u(x,y,t) = \nu\Delta u(x,y,t), \ (x,y) \in \Omega,$$

where $\Omega = [-1,1]^2$. We are interested in learning the solution operator $S_{\tau_1,...,\tau_n} : u_0 \mapsto \{u(\cdot, \tau_1), ..., u(\cdot, \tau_n)\}$. In the dataset, the initial condition $u_0(x)$ is generated using a Gaussian random field according to $u_0 \sim \mathcal{N}(0, 16(\Delta + 16I)^{-2})$ with Neumann BCs, and $\nu$ is fixed to $0.001$. The dataset consists of 1000 training instance and 100 test instance.

During the training, parameters for SPFNO are chosen as $W = 24$ and $K = 16$, and all models are trained for 3000 epochs, following the instruction of Liu et al. (2022b). The bandwidth $b$ is fixed to $4$.

### B.3 Example 3 and 4: PDE problems from PDEBench dataset

We set the parameters for SPFNO and FNO as $W = 24$ and $K = 24$, and all models are trained for $500$ epochs, following the instruction of Takamoto et al. (2022). The bandwidth $b$ is fixed to $1$.

### B.4 Additional example: PDE problems from BOON dataset

We choose the same parameters for both SPFNO and BOON models as those in the paper (Saad et al., 2023): $W = 64$ and $K = 32$ for 1D problem, and $W = 20$ and $K = 16$ for 2D time-varying problems. Note that each Fourier basis function is equivalent to two real function. The SPFNO is trained for $500$ epochs with $b$ fixed to $1$.

