# OpenReview forum: "SPFNO: spectral operator learning for PDEs with Dirichlet and Neumann boundary conditions"
_ICLR.cc/2024/Conference — ICLR 2024 Conference Withdrawn Submission_

### Official Review · Reviewer_81kc · 2023-10-29

**Soundness:** 2 fair
**Presentation:** 1 poor
**Contribution:** 2 fair
**Rating:** 3
**Confidence:** 5

**Summary:**

This paper proposes SPFNO, a so-called spectral operator learning method to solve PDEs with Dirichlet and Neumann boundary conditions on uniform grids. SPFNO leverages trigonometric polynomials and sine/cosine transforms to precisely satisfy boundary conditions.

**Strengths:**

1. Explore methods on different Dirichlet/Neumann BCs to further improve the ability of neural operator series work.
2. Provides fast O(NlogN) transforms for spatial differentiation using spectral methods.

**Weaknesses:**

1. The proposed Semi-Periodic FNO appears to be overly simplistic and lacks in-depth analysis and exploration. For instance, the absence of alternative basis selection limits the thorough examination of its capabilities. As a result, the paper appears more akin to a technical report rather than a scientific research article.

2. The experiments are rough without in-depth analysis. For example, the paper lacks ablation studies to validate design choices like bandwidth of learnable matrices, no analysis provided on how accuracy scales with problem size and dataset size, and it does not discuss performance on other complex boundary conditions like Robin or mixed BCs. The most important thing is that it does not compare with other BC satisfying neural operator methods and claim it is sota which cannot convince me.

3.  There is no evidence that satisfying BCs improves generalization and robustness.

4. Limited hyperparameter tuning details are provided for model architecture and training.

**Questions:**

1. Can you expand studies to other boundary conditions like Robin, mixed, time-dependent, etc
2. Can you add ablation studies on model architecture choices
3. Can you demonstrate how accuracy changes with dataset size and problem dimensionality
4. Can you compare with other BC satisfying operator learning methods
5. Can you show if satisfying BCs improves robustness through noise tests, outlier evaluation, etc
6. Can you provide more details on model tuning, architecture search, regularization techniques used
7. Can you enhance readability by adding more architectural and algorithmic details to sections

---

> ### Author Response · Authors · 2023-11-21
>
> Thank you for your insightful discussion and suggestions, which have been greatly beneficial to our paper.
>
> **weekness**
>
> W1 **alternative basis**: From the perspective of underlying bases, many recent existing work following FNO, such as MWT-NO (NIPS 2021), WNO (CMAME 2023), SFNO (ICML 2023), SNO (Vladimir & Ivan), and the SOL models(OPNO), are all focus on exploring neural operators induced by different bases and they share the same pseudospectral backbone: learning linear transforms in frequency domain and activating in physical domain. Crucially, the SOL architecture emphasizes that, as in spectral methods, matching with BCs is a criterion for selecting the underlying basis. This is not merely mathematical intuition; experimental results in solving multiple equations also prove it, especially on third-party and publicly available PDEBench and BOON datasets.
>
> W2 **analysis**: We now add the comparisons with BOON, another BC-satisfying neural operator, in cases of Dirichlet and Neumann BCs. In addition, the influence of bandwidth of learnable matrices of SPFNO and FNO can be found in Fig.4; the applications on incorrect BCs contradicts the mathematical principles of the model; and the method of selecting subsets from the training set could potentially lead to an unfair comparison.
>
> W3 **generalization and robustness**: Learning to solve PDEs is quite different from other deep learning scenarios like image recognition and natural language processing, and we would like to leave the detailed discussion in our response to Q5.
>
> W4 **hyperparameter details**: We agree that although the complete code for each experiment with fixed random seed are given on the Github pages, so the results are fully reproducible, it would be helpful for reader to be informed of the hyperparameter details. We add more details in the appendix. Thanks for the advice!
>
> **Questions**:
>
> Q1. **different BC**: We add the experiment for time-dependent BCs in 3.5.1 on BOON dataset. For mixed BCs, actually we had developed and uploaded to Github the code (models.WSWANO using a “1/4 trigonometric basis”). We have tested it on pipe dataset of GEO-FNO with good results, but we found a very small number of data in current dataset are incorrect. After communication, the authors have fixed the bugs, and the application on mixed BCs is to be tested after the updated dataset is released.
>
> The sine basis for Robin BCs is given in [1] but needs a major modification to the neural operator. Therefore, we may leave this part for future research.
>
> [1] Ilic, Milos, et al. "Numerical approximation of a fractional-in-space diffusion equation, I." Fractional Calculus and Applied Analysis 8.3 (2005): 323-341.
>
> Q2. **ablation studies**: Mathematical intuition tells us that satisfying boundary conditions is of great importance to neural operators. This paper aims to validate this hypothesis. In the comparison between SPFNO and other operators like FNO, we have already avoided various neural network tuning tricks to ensure that factors are controlled and comparison is fair.
>
> Q3. **problem dimensionality**: Even if equations of different dimensions have the same form and have chosen the same parameters, the evolution of the system they describe is completely different. Therefore, we’re afraid the performance on different dimensions are not comparable. Additionally, selecting a subset of the training set for model comparison may not be fair to different models.
>
> Q4. **comparison with other BC-satisfying operator**: Thanks for the advice and we have compared SPFNO with the BC-satisfying BOON model now. We did not fully reproduce BOON, especially the code for the 2D Neumann boundary of BOON has not been made public yet, and the implementation of this part is quite complex. Therefore, in the previous experiments, we can only made a comparison in Experiment 1. Fortunately, we also made comparisons on the Dirichlet and Neumann BCs in the BOON dataset.
>
> Q5. The problem is that robustness and generalization are not well-defined for learning PDEs. In image recognition, altering a small number of pixels or adding noise doesn't change the label, which is the concern in the study of adversarial examples. However, modifying the input of a PDE will certainly affect the output. Learning a damped model for an ill-conditioned problem doesn't mean it's better.
> The dataset in image recognition is influenced by the collection process, while the input for PDEs is randomly generated. Furthermore, perhaps evaluating on a grid different from the training grid could be a measure of model generalization, please see section 3.5.2 that we add.
>
> Q6. We have added the details of the model in Appendix B.
>
> Q7. Thanks for the suggestion. We reorganized Fig. 2 to add more architectural details.

---

### Official Review · Reviewer_YuJS · 2023-10-30

**Soundness:** 1 poor
**Presentation:** 1 poor
**Contribution:** 1 poor
**Rating:** 1
**Confidence:** 4

**Summary:**

This paper considers the problem of learning solution operators associated with partial differential equations (PDEs) with deep learning. The authors propose a new architecture, in a similar spirit of the popular Fourier neural operator, based on the discrete sine and cosine transform to enforce Neumann or Dirichlet boundary conditions strongly in operator learning problems, and perform convolutions in the feature space using Discrete Cosine/Sine transforms (DCT/DST). Then, the authors perform deep learning experiments and compare their approach using FNO and other architectures.

**Strengths:**

- The FNO enforces periodic boundary conditions strongly but Dirichlet or Neumann are more common in practical applications and are enforced weakly by current neural operator architectures.
- The numerical experiments show that the authors' approach achieve close to machine precision error on the boundary conditions, which is an improvement over the FNO.

**Weaknesses:**

- The paper is not well-written and contains many typos and spelling mistakes.
- The relation between this work and the previous work by Liu (2022b) is very confusing. The authors mention that their work is an improvement over Liu (2022b): the difference is that Liu et al's approach requires Gaussian grid to evaluate the neural operator while the present paper allows for uniform grid. First, this seems to be a minor contribution. Secondly, Subsection 2.1 and Fig. 2 are copied from Liu 2022b but not references as such.
- The SPFNO approach is not well introduced.
- The experimental results also contain several drawbacks. First, the experiment setup is not provided, and the number of parameters between models is not comparable. In Example 2, OPNO (Liu, 2022b) outperforms the paper's approach, but is not evaluated in examples 3 and 4. The training time for FNO is much faster than SPFNO (factor 1.83 and 3.58 for experiments 3/4, respectively).

**Questions:**

- What is the motivation for using uniform grid as it might lead to aliasing errors?
- 2nd paragraph of p.2 in bold: which fast transform?
- There is no description of Fig. 1 in p.2.
- First paragraph of Section 2.1: "respecting boundary conditions exactly" does not make sense, it should be strongly.
- Last sentence of Section 2.1: "probably sparse structure": what does that mean?
- Theorem 2.1: there is no assumption on the function f (i.e. its function space).
- Theorem 2.2: the statement is not clear, what does strictly satisfy the boundary conditions mean?
- Appendix A.1: The proof of Thm. 2.1 is not rigorous: where does Eq.(11) comes from? The authors should state that it's the representation of f in an orthonormal basis of L^2. The coefficients a_m, b_n should be introduced/defined. Corollary A.1 comes without a proof.

---

> ### Author Response · Authors · 2023-11-21
>
> Thanks for your review and helpful suggestions!
>
> **Weakness**
>
> The number of parameter is actually comparable. We have selected the same modes, which are one of the most important parameters for spectral methods, as referenced in [1]. Besides we choose the same width W and depth L, and a bandwidth greater than 1 does not improve the accuracy of FNO, as shown in Figure 4. We have also tried increasing the width and modes of FNO. However, simply increasing parameters in this way cannot equate to the improvements brought about by satisfying boundary conditions.
> [1] John P Boyd. The choice of spectral functions on a sphere for boundary and eigenvalue problems: A comparison of chebyshev, fourier and associated legendre expansions. Monthly Weather Review,
> As reviewer wNAD pointed out, the reason for the faster efficiency of FNO is due to PyTorch's FFT implementation, while the butterfly operation used in our program's DCT/DST has not yet been optimized, which results in a constant multiple computational overhead.
> 106(8):1184–1191, 1978.
> The reason we did not evaluate the model in Experiments 3 and 4 has been pointed out in the text: using OPNO on a uniform grid is incorrect.
>
> **Questions**
>
> Q1. **Why uniform grid**:  Besides the reasons mentioned in author response, there don't seem to be any noticeable aliasing errors in the experiments. In fact, the truncation performed by neural operators in the frequency domain on the input is very similar to the filter used by Philips and Orszag, which eliminated the aliasing. See equation (11.13) in [2].
> [2] John P Boyd. Chebyshev and Fourier spectral methods. Courier Corporation, 2001.
>
> Q2. fast discrete sine/cosine transform.
>
> Q3. Description can be found in the last paragraph of section 1.
>
> Q4. Thanks for figuring it out and we have correct it.
>
> Q5. For most of FNO-like neural operators listed in Tab. 1 the learnable matrix is diagonal, but for OPNO it is tridiagonal for most of the cases. And for multi-channel IAE-Net, the matrix is of high-rank and generated by a neural network.
>
> Q6. Thank you for pointing out and in the proof we only need $f \in C[0, 1]$.
>
> Q7. The output of arbitrarily initialized SPFNOs satisfy corresponding BCs.
>
> Q8. We have refined the proof for the theorems.

---

### Official Review · Reviewer_wNAD · 2023-10-31

**Soundness:** 2 fair
**Presentation:** 3 good
**Contribution:** 2 fair
**Rating:** 5
**Confidence:** 4

**Summary:**

The authors propose SPFNO, which is a spectral operator leaning (SOL) method that enforces boundary conditions within Neural Operator methods.  By incorporating boundary conditions and techniques from numerical analysis, the proposed method is shown to be more accurate than FNO and other data-driven methods on a range of experiments but is not compared to BOON which also enforces boundary conditions in Neural Operators to obtain higher accuracy using an alternative approach. In particular, the authors propose to use a sine basis function for Dirichlet BC and cosine basis function for Neumann BC, which enforces the constraint on the derivative of the solution by using odd and even extensions, respectively.

**Strengths:**

- It is good to incorporate ideas from numerical methods such as spectral methods and boundary conditions to improve the accuracy of pure data-driven methods for PDEs.
- Nice mention of stability, numerical convergence and matrix sparsity from numerical methods that can be advantageous to leverage within DL methods.
- It is nice to see that as in BOON (Saad et. al, ICLR 2023) enforcing the boundary conditions improves the accuracy of the model.
- Good overview of numerical methods in the introduction.
- Good that the method does not break the desired resolution invariance property of neural operators.
- Nice test case on challenging shock problem with 1D Burgers equation where most DL methods are highly oscillatory. It looks like in Figure 1b though that the resulting solution profile from SPFNO still has damped oscillations, violating the maximum principle and monotonicity of the true solution.
- Nice visualization of the plots but Figure 1e is much too small.
- Nice and simple idea to use a sine basis for Dirichlet BC and cosine basis for Neumann BC.
- It is nice that the authors provide corresponding theory for their method and should that the BC are strictly satisfied.
- Interesting finding that it is easier to learn Dirichlet than Neumann BC which involved derivatives. The error in approximating the derivative also comes into the schem.

**Weaknesses:**

- It is better to first motivate the PDE problem definition and application in science and engineering before diving into Neural Operators specifically.
- There are missing references to other state-of-the-art SciML classes of methods other than Neural Operators including MeshGraphNets (Plaff et. al, ICLR 2021) and PINNs (Raissi et. al, 2019).
- One major weakness is that the related BOON method from ICLR 2023 "Guiding continuous operator learning through Physics-based boundary constraints" by Saad et. al, is not cited or compared to. This is the first method to enforce boundary conditions in arbitrary Neural Operators, resulting in a 2x-20x improvement.
- Multi-wavelet Neural Operator (Gupta et. al, ICLR 2021) is also not compared to in the evaluations.
- It is also incorrectly stated in the intro that most of the public datasets have periodic BC. Please find the datasets from BOON with Neumann and Dirichlet BC in the open-source repo https://github.com/amazon-science/boon.
- The cited multi-wavelet NO (Gupta et. al, NeurIPS 2021) also leverages orthogonal polynomials as well as the state-of-the-art MeshGraphNets (Plaff et. al, ICLR 2021)
- The wide applicability of FNO is more due to the ease of the use and efficiency of the pytorch FFT implementation rather than its "straightforward training process".
- Sine and cosine bases may still not be ideal bases for hyperbolic conservation laws with shocks.
- The authors mention Robin BC but do not show experiments with it.
- The architecture sketch in Figure 2 can be moved to an Appendix since the architecture is very similar to FNO just with using separate sin/cos bases rather than Fourier, which also limits the novelty.
- Page 6 is sparsely filled with results and no text. There is room for additional analysis and more PDE experiments including see the lid-driven cavity flow Navier-Stokes Dirichlet BC test case from BOON https://github.com/amazon-science/boon.
- The results analysis is missing and without a discussion it is very difficult to interpret Figure 3.
- I'd like to see the proof for Neumann BC since that involved the derivative. Also the proof for Corollary A.1 is missing.

Minor
- Importance is spelled incorrectly in the abstract.
- The second sentence in the introduction is extremely long and should be broken up at the applications of FNO to weather forecasting.
- "after acceptance" rather than "after accepted" in last paragraph of the intro.
- Section 3 should be plural "Numerical Experiments"
- Commas missing after some equation
- "verry" close typo at the end of page 5.
- In 3.4, it should read "the task is to learn"
- Conclusion should read, "Although the paper focuses"
- "yeilds" typo in A.1

**Questions:**

1. Since sine or cosine basis is used, why are the oscillations damped with SPFNO on problems with shocks, i.e., Burger's equation here?
2. How can this method be extended to non-uniform grids?
3. Does LSM beat MeshGraphNets? The wording on this method being state-of-the-art is quite strong since it also doesn't compare to MeshGraphNets (Plaff et. al, ICLR 2021).
4. There is a lot of comparison to Liu et. al, 2022b (OPNO), what is the differences in terms of novelty since Liu et. al, 2022b also satisfies the boundary conditions.
5. In particular, in table 1a, it looks like OPNO wins in a majority of cases over the proposed CosNO for Neumann Bc - why is this?
6. The resolution $n$ is quite fine in Table 1 up to 4096 nodes for 1D Burgers which is extremely fine and not required for numerical methods in 1D. They can be accurate for $n=64, 128, 256$ and would have already converged to very low error on order of less than half machine precision $<1e-8$ by 4096. Why are such fine grids considered here and why is the error is still quite large on the order of $1e-2$?
7. Also then in 2D the resolution is much coarser - is there due to the computational issues moving from NO in 1D to 2D, which is the only reason DL methods should be considered over numerical methods. If both suffer the curse of dimensionality, its hard to see the use of DL methods.
8. Is $n$ supposed to be $N$?
9. It is mentioned that example 3.3 is from PDEBench and the baselines are taken from there - is it a fair comparison with the same hyperparameters and number of parameters for reproducibility.
10. Why does LSM perform better in Table 4 than Sin-NO in the worst case error?

---

> ### Author Response · Authors · 2023-11-21
>
> Thank you very much for your detailed suggestions. We have made adjustments to the image size, corrected typos, and so on.
>
> **Weaknesses**
>
> W1 **Introduction for the PDEs**: That's a good advice and we agree that this article should be targeted at a broader audience. We have add introduction and motivation of the PDEs problem at the beginning of our article.
>
> W2 **Introduction for DL models**: Similarly, Similarly, we have reorganized the introduction to the deep learning methods for solving PDEs in the Section 1.
>
> W3 **comparison with BOON**: Thank you for pointing out our oversight. The comparison with the BOON model is now one of the focuses of our paper. And now we are still trying to add comparison on more problems.
>
> W4 **comparison with MWT-NO**: We have conducted preliminary experiments on representative Examples 1 and 4, which include both 1D and 2D cases, as well as Dirichlet and Neumann BCs. In Example 1, the errors of MWT-NO are (0.039,0.036,0.033) for N=(256,1024,4096), which are better than any non-neural operators. And in Example 4, the error of MWT-NO is 0.00568, which is better than FNO. We are still working on supplementing numerical experiments, but I'm afraid the priority is relatively low.
>
> W5 **dataset**: We are so happy to know this! For now we have conducted experiment on dataset of both Dirichlet and Neumann BCs from BOON, as well as single/multi-steps.
>
> W6 **orthogonal polynomials**: Thank you for your supplement.
>
> W7 **introduction of FNO**: We have reorganized the introduction. The meaning of this sentence was not clear enough, so we have removed them.
>
> W8 **Sine and cosine bases**: We are interested in the cases but maybe we need practical experiments to verify this.
>
> W9 **Robin BC**: For most of the cases the spectral method of sine/cosine are applied on Dirichlet or Neumann BCs, but [1] gives the explicit form of the sine basis for mixed Dirichlet-Robin BCs in Example 2. However, its induced neural operator needs a major modification from existing ones (compared with the OPNO for Robin BCs). So I guess it is out of scope of this paper and for now we recommend OPNO for solving PDEs of Robin BCs.
> [1] Ilic, Milos, et al. "Numerical approximation of a fractional-in-space diffusion equation, I." Fractional Calculus and Applied Analysis 8.3 (2005): 323-341.
>
> W10 **sketch map**: Thanks for your advice and we reorganized the sketch map to describe the structure of SPFNO more clearly. The sine/cosine of half period is actually not merely half of Fourier, which is demonstrated from the proof of the theorem 2.1.
>
> W11 **more PDE experiment**: Although we have added multiple experiments, we would like to discuss the lid-driven cavity experiment. For spectral method, it is more common to compute a regularized driven cavity instead to properly evaluate the accuracy, such as [2,3]. Similarly, we are uncertain about how to correctly apply discontinuous boundary conditions to the SPFNO. The lid-driven cavity is physically unrealistic but frequently used as a benchmark problem for academic usage [4]. So I wonder whether learning this problem could approximate real physical phenomena, or merely capturing the features of some numerical solvers. We are interested in this problem and looking forward to continuing the discussion in future.
>
> [2] Shen, Jie. "Numerical simulation of the regularized driven cavity flows at high Reynolds numbers." Computer methods in applied mechanics and engineering 80.1-3 (1990): 273-280.
>
> [3] Shen, Jie. "Hopf bifurcation of the unsteady regularized driven cavity flow." Journal of Computational Physics 95.1 (1991): 228-245.
>
> [4] Botella, O., and R. Peyret. "Benchmark spectral results on the lid-driven cavity flow." Computers & Fluids 27.4 (1998): 421-433.
>
> W12 **Fig 3**: The Fig. 3 can serve as an ablation study, as mentioned by the Reviewer 81kc. We want to show that the reason of better performance of SPFNO is due to the BC-satisfying property rather than a wider bandwidth, and it also illustrates the influence of bandwidth for SPFNO. We add further discussion in Appendix B.
>
> W13 **proof for Neumann BC**: We have refined the proof now in Appendix A.
>
> **Minor**
>
> Thank you again for your detailed suggestions and we are sorry for the bugs in our first draft.

---

> > ### Author Response · Authors · 2023-11-21
> >
> > **Questions:**
> >
> > Q1. Perhaps we have not fully understood your meaning. The viscous term prevents the occurrence of shocks in the Burgers' equation.
> >
> > Q2. Our answer is divided into two parts. Firstly and in a traditional way, the sine/cosine transform can also be carried out on a non-uniform grids, and the inverse transform is merely sampling on the output function series. The problem is that the model may lost the fast transform algorithm and reduce the efficiency. Secondly and in a neural-network way, we may first maps the input function to a latent space and then apply the SPFNO, following the idea of Geo-FNO.
> >
> > Q3. The results of LSM is quite admirable for a non-neural operator model. I’m afraid comparing MeshGraphNets with LSM might go beyond the scope of this paper, so we have made minor modifications to the introduction of LSM (most accurate non-neural-operator model for PDEs known to us).
> >
> > Q4. Due to the difference in grid types, they are suitable for different scenarios. Since it is incorrect to use the interpolation polynomial on a uniform grid as the input for OPNO, which leads to Runge phenomenon, OPNO is hard to be applied on uniform grids. Before the proposed SPFNO, the SOL architecture has not been compared with other models on third-party datasets. So from an empirical standpoint, SPFNO significantly enhances the credibility of our SOL architectures.
> >
> > Q5. Actually with bandwidth=7, the performance of CosNO is slightly better than OPNO for 1D Burgers’ equation, but it requires more resources. See Figure 4. Besides, the worst-case error of SPFNO is lower for 1D-Burgers, which is also an important performance metric.
> >
> > Q6. The reason is that in Example 1 and 2 the initial condition u0 is generated by a Gaussian random field. It introduces degrees of freedom on the same order of magnitude as N. The input functions in these experiments may have relatively poor smoothness, so it is necessary to choose a sufficiently large N, even though high-precision spectral methods were used to generate this dataset.
> >
> > Achieving high performance on these datasets may require a larger amount of data, similar to the Heat-Robin experiment in the OPNO paper. We also have a related discussion in section 3.5.1.
> >
> > Q7. Not really, the reason of lower resolution of 2D cases is mainly because generating datasets using numerical methods on fine grids is too slow, especially for non-periodic cases. The lack of high-quality datasets can be a bottleneck issue for the development of deep learning methods for PDEs.
> >
> > Q8. You are right. Sorry for the mistake.
> >
> > Q9. Yes. The FNO and SPFNO are of the same modes and width, and trained for the same 500 epochs. The parameters required for FNO, SPFNO, Unet and LSM are 1.3m, 1.4m, 7.8m, 1.2m, respectively, so the comparison is fair.
> >
> > Q10. The primary factor contributing to the SPFNO's worst error compared to LSM lies in the SOL method's spectral structure, which also affects the FNO. The input functions of the PDEBench-Darcy dataset are discontinuous, requiring more frequencies. Due to the reference to the hyperparameter selection in  PDEBench, in the experiments 4, SPFNO requires approximately 12% of the parameters of LSM. Increasing the number of modes and width can enhance the performance of SPFNO.

---

### Official Review · Reviewer_FprA · 2023-11-11

**Soundness:** 1 poor
**Presentation:** 2 fair
**Contribution:** 1 poor
**Rating:** 3
**Confidence:** 5

**Summary:**

In this paper authors propose a new Spectral Operator Learning (SOL) framework for solving partial differential equations (PDEs) that involve Dirichlet or Neumann Boundary Conditions. To enforce non-periodic boundary constraints, the proposed method utilizes the discrete sine and cosine transform. The  proposed architecture was evaluated on a variety of PDE problems and compared against a few baselines including original FNO, Unet, Latent Spectral Models (LSM), and Orthogonal Polynomial Neural Operator (OPNP). The results indicate precise satisfaction of boundary constraints and higher overall accuracy for some test cases.

**Strengths:**

- It is great to see that there are researchers in the field of SciML who believe a pure data-driven approach with the hope of learning all physics constraints "implicitly" from data is NOT enough, and it is important to use readily available physics information in addition to input datasets.
- Authors considered a good variety of PDE problems with different applications and level of difficulty, and it is nice to see their approach is capable of enforcing Dirichlet and Neumann boundary constraints at machine precision level.
- It is also nice that authors shared their codebase and data, which are useful for benchmarking with other researchers.

**Weaknesses:**

- My main concern is about the originality of this work. The idea of using spectral methods/orthogonal polynomial transformation in neural networks for solving PDEs is not really a novel idea. Authors already mentioned two important methods, LSM and OPNO, which utilize the power of spectral methods, and they seem to have competitive overall performances. To my understanding, the only update in this work is the use of different orthogonal transformation (discrete Cosine or Sine) while keeping the rest of configuration the same as FNO.
- I would like to challenge the statement of "neural operators trained on coarse grid can predict results on finer mesh". Authors should keep in mind that neural operators are purely data-driven methods and may not perform well on finer grids if they are not given the chance to learn the pattern in sub-scale grids. Also it is not scientifically appropriate to mention something (even from other papers) and not to provide numerical evidence supporting the statement.
- I found that there are a few closely related papers missing from the references. Just to mention few examples:
   - (BOON) Guiding continuous operator learning through Physics-based boundary constraints (https://arxiv.org/pdf/2212.07477.pdf)
  - Neural Q-learning for solving PDEs ( https://www.jmlr.org/papers/volume24/22-1075/22-1075.pdf)
  - Physics-Embedded Neural Networks: Graph Neural PDE Solvers with Mixed Boundary Conditions (https://proceedings.neurips.cc/paper_files/paper/2022/file/93476ae409ae3246e22a9d4b931f84ed-Paper-Conference.pdf)
  - Enforcing Dirichlet boundary conditions in physics-informed neural networks and variational physics-informed neural networks (https://www.sciencedirect.com/science/article/pii/S2405844023060280)
  - and more!
It would be nice that authors make sure that they have done a comprehensive literature review and this will further improve the credibility of the research conducted. For example, the first reference mentioned above actually enforces boundary conditions by systematically adding additional layer to a neural operator architecture, and could be a great baseline for comparison.
- Lastly, the presentation of some sections of the paper could have been further improved. For example, the evaluation procedure is not described clearly. It is not clear what resolutions considered during training and what is considered during evaluation, whether the evaluations are in-domain predictions (seen at training time) or out of domain, what is the number of replicates of the same problem and whether they reported ensemble results or the results come from one single run, etc. These details are important for future benchmarking with other researchers and authors should

**Questions:**

As addressed briefly above in "weaknesses" section, I wonder if the numerical results reported came from one single run of experiment or it is the ensemble of multiple runs? It is important that authors conduct multiple experiments of the same problem with random seeds and report both mean and standard deviation to better illustrate the robustness of their results.
- For majority of the results one can see that some baselines such as OPNO for Burgers equation, and LSM for Darcey flow can perform similarly or even better than the proposed method. Could authors provide more detailed discussions on the results and comment on why their method may not result in the most accurate predictions among all baselines. This shows that although boundary constraints are satisfied precisely then there are other sources of errors in their method. What are potential sources?
- Did authors conduct any experiment to investigate the "resolution invariance" property of neural operators, because they list it as one of their contributions? For this, it would be nice to train their model on coarse mesh and evaluate on finer mesh, and vice versa and see how model predictions look like for each scenario.

**Details Of Ethics Concerns:**

No ethics concern

---

> ### Author Response · Authors · 2023-11-21
>
> Thanks for your insightful review and valuable suggestions.
>
> **Weakness**
>
> W1. **Originality**: Thanks for the discussion. Except choosing the correct bases to BCs, we deliberately avoid extra modifications to the FNO backbone and tuning tricks, in order to validate the importance of BC-satisfying property. For spectral method, the marching between BCs and bases are the most crucial issue. A spectral method is usually specifically designed for particular BCs and geometry (Fourier for periodic BCs, Spherical Harmonics for sphere, and Hermite for unbounded domain, etc.). Correctly choosing the bases can automatically introduce physical information of BCs into the model, while incorrect selection may introduce implicit artificial constraints.
> Besides, both SPFNO and OPNO are SOL architectures. One of the motivations behind SPFNO is to address the issue of OPNO 's lack of comparison with other models on third-party datasets, as we mentioned in author response.
>
> W2. **Invariance to discretization**: We agree with your viewpoint. We have added experiments in Section 3.5.2 where the training and prediction grids of the model are different. We would like to add that the invariance to discretization can be used to dramatically accelerate pre-training using coarser grids in large-scaled tasks, such as global weather prediction.
>
> W3. **Related works**: Thank you for your suggestion! The comparison with the BOON model is now one of the focuses of our paper. And we have improved the introduction of the relevant field work in Sections 1 and 2.1, following your advice. Compared to neural operators, the research on improved PINN-like models that can strictly satisfy BCs is quite extensive, but these techniques generally cannot be directly extended to neural operators.
>
> W4. **presentation**: We have provided these additional information in our paper. Except for Section 3.5.2, both training and testing are conducted on the same grid. We set the random seed to 0 in training so generally only perform one single run.
>
> **Questions**
>
> Q1 **multiple runs**: We agree that multiple runs would be meaningful. However, some experiment like autoregressive training in Experiment 3 is quite costly, multiple runs is beyond our computational resources. Therefore, we decided to set the seed to 0 for all models for fairness. It is a very common decision in the paper of neural operators, and as [Choose a transformer: Fourier or galerkin, NIPS 2021] said,  “the impact due to the choice of seeds for our models is empirically minimal.” One possible reason is that for the trained neural operator model, most of the energy is concentrated in the low-frequency part of the learnable spectral matrix A_l. So neural operators with structures similar to FNO rarely fall into local minima during training, and the impact of different random seed is minimal once the model is well-trained.
>
> Q2. The primary factor contributing to the SPFNO's worst error compared to LSM lies in the SOL method's spectral structure, which also affects the FNO. The input functions of the PDEBench-Darcy dataset are discontinuous, requiring more frequencies of neural operators. Due to the reference to the hyperparameter selection in PDEBench, in the experiments 4, SPFNO requires approximately 12% of the parameters of LSM. Increasing the number of modes and width can enhance the performance of SPFNO.
>
> Q3. As mentioned in the reply to W2, we added related experiments in Section 3.5.2.

---

### Author Response · Authors · 2023-11-21

We appreciate the questions and suggestions raised by the reviewers, which have greatly enhanced the presentation of the article. First we would like to briefly address the potential concerns that multiple reviewers might have.

1.**Comparison with BOON**. The absence of comparisons to existing BC-satisfying models that is an oversight that should be avoided. We add the comparison with BOON (ICLR 2023) in Sec. 3.1 and 3.5.1. Because SPFNO naturally satisfies the BCs without the need for corrections, it exhibits higher computational efficiency (refer to Tab. 6), and outperforms BOONs in accuracy in almost all cases. Furthermore, in the experiment of Burgers’ equation with Neumann BCs, the accuracy of BOON surpassed all the other non-BC-satisfying models, including FNO, which is consistent with the BC-satisfying criterion we proposed.

2.**Novelty, and relationship with OPNO**. In an academic report on the SOL model, some audience members believed that the SOL model lacked comparisons with other models on third-party datasets, which undermines the persuasiveness of its results, especially the necessity of strictly satisfying BCs. However, although the Gaussian grids are widely used in computational mathematics, it is unrealistic to expect other scholars to provide solutions on them in their datasets, and using polynomial interpolation on uniform grids will cause severe oscillations from Runge phenomenon. The SPFNO tackle the technical issue and make comparison on datasets from PDEBench and BOON. The results of this paper are crucial for the acceptance of SOL methods within the scientific community.
The performance of neural operators with different bases has been the main theme of many recent papers, to list a few, MWT-NO(NIPS 2021), WNO (CMAME 2023), IAE-Net (JMLR 2022), SFNO(ICML 2023), etc. Importantly, our paper clarifies that the correct basis and spectral transformation should be chosen according to the boundary conditions (BCs); in addition, most transformations are slow, such as those for Legendre polynomials, spherical harmonic functions, etc., while the DCT and DST used in this paper can use fast algorithms.

3. **presentation**: We have expanded the introduction to the background and the SPFNO architecture, and further refined the proof of the theorems.